# Black Bean Hulls as a Byproduct of an Extraction Process to Enhance Nutraceutical and Glycemic-Related Properties of Nixtamalized Maize Tostadas

**DOI:** 10.3390/foods12091915

**Published:** 2023-05-08

**Authors:** Lesly Xiomara Machado-Velarde, Juan Pablo Dávila-Vega, Janet Gutiérrez-Uribe, Johanan Espinosa-Ramírez, Mariana Martínez-Ávila, Daniel Guajardo-Flores, Cristina Chuck-Hernández

**Affiliations:** 1Tecnologico de Monterrey, School of Engineering and Sciences, Ave. Eugenio Garza Sada 2501, Monterrey 64849, NL, Mexico; 2Tecnologico de Monterrey, Institute for Obesity Research, Ave. Eugenio Garza Sada 2501, Monterrey 64849, NL, Mexico

**Keywords:** black bean hulls, maize tostadas, anthocyanins, dietary fiber, functional food

## Abstract

Black bean hulls (BBH) are rich in phenolic compounds, such as anthocyanins, which can be incorporated into common staple foods such as maize tostadas, enhancing the nutraceutical properties of these products. This study incorporates black bean hulls to produce nixtamalized maize tostadas with nutraceutical properties. Nixtamalized corn flour (NCF) and black bean hulls (BBH) were characterized in terms of protein, fat, crude and dietary fiber, anthocyanin concentration, and different starch fractions. NCF and BBH depicted 53.7 and 16.8% of total digestible starch (TDS), respectively, and 1.2 and 7.6% of resistant starch (RS), in the same order. BBH was incorporated into nixtamalized flour at 10, 15, and 20% *w*/*w*, and the resulting dough was thermo-mechanically characterized. Tostadas with BBH had higher protein, dietary fiber, and anthocyanin concentrations. Enriched tostadas did not show significant changes in texture or other sensory characteristics. However, a reduction in total digestible starch (61.97 up to 59.07%), an increase in resistant starch (0.46 to 2.3% from control tostadas to 20% BBH tostadas), and a reduction in the predicted glycemic index (52 to 49), among other parameters, indicated that BBH is a suitable alternative for developing nutraceutical food products.

## 1. Introduction

Multiple countries have adopted circular economy concepts, a production and consumption model to reduce waste and optimize the use of resources in all supply chains. The circular economy involves designing and implementing novel alternatives considering environmental sustainability and economic and social development [1]. According to the FAO, around 160 million tons of food waste is generated yearly in the production sector [2]. Waste of legumes constitutes a significant portion of the pulses produced, in which up to 25% of the beans are discarded as residue or waste during their processing [3]. In 2021, the common bean’s (*Phaseolus vulgaris* L.) production was 27.7 million tons, making it one of the most consumed legumes worldwide [4]. The bean residues include the hull, powder, and shriveled seeds, which are mainly used as animal feed; however, they could also be a valuable source of protein, minerals, and bioactive components for producing nutraceutical products.

Black beans (*Phaseolus vulgaris* L.) are pulse seeds widely consumed due to their proven health benefits, such as glycemic modulation, cardioprotective function, antioxidant, and anti-inflammatory effects, among others [5]. Therefore, recent studies have focused on developing black bean flour with better nutritional properties to make affordable food products [6,7,8]. However, while producing these high-quality flours from black beans, waste is generated mainly from the hulls. The hulls represent around 11% of beans and contain antinutrients, such as protease inhibitors, lectins, and phytic acid that reduces the absorption of protein and minerals during ingestion. Additionally, these materials also contain large amounts of nutritionally desirable molecules like saponins and phenolic compounds such as flavonoids and anthocyanins [9,10], with important antioxidant, anti-inflammatory, antidiabetic, and hypertensive activities [11,12]. In fact, some companies use black bean hulls to produce food supplements, obtaining spent materials with residual anthocyanin and high fiber contents. By this means, novel strategies must be addressed to use the spent black bean hull as a valuable ingredient to reduce food waste and promote the production of functional and high-consumption products.

In Mexico and some Latin American countries, nixtamalization, an ancient and pre-Hispanic technique, is used to produce maize dough and staples such as tortillas, tortilla chips, taco shells, tostadas, and tamales, among others. Briefly, maize kernels are lime-cooked with Ca(OH)_2_, followed by an overnight steeping process with a subsequent rinsing of the kernels with tap water, and then milled into a malleable dough [13]. However, only in Mexico were most of the 27.4 million tons of maize produced in 2020 transformed through nixtamalization into those products [13], including tostadas, a flat, crispy, or toast tortilla made of nixtamalized corn, with low water activity and vast consumption with different toppings. This study incorporates black bean hull residues to produce tostadas with bioactive compounds, increased protein digestibility, and a moderate glycemic index.

## 2. Materials and Methods

### 2.1. Raw Materials

Nixtamalized corn flour (NCF) was obtained from local Maseca^®^ and Minsa^®^ suppliers, and black bean hulls (BBH) were obtained from a process of anthocyanin extraction for supplement elaboration from JASEDA industries (Monterrey, Mexico). BBH decortication was undertaken according to the method previously reported by [14]. The sample was cleaned with a flannel and then soaked in distilled water in a bean/water ratio of 100:1 (*w*/*v*) at room temperature for 24 h. The conditioned black beans were dried at 60 °C for 6 h in an oven (Electrolux, EOB31003X, Stockholm, Sweden). Later, the seed hulls were removed with a mechanical seed decorticator (Square D, SC-DGE 4364, Goyum Group, Punjab, India) for 90 s, and separated them using a sieve (1.00 mm). Additionally, to obtain the nixtamalized corn flour (NCF), flours with different particle sizes were mixed:“Minsa” coarse flour, “Minsa” fine flour, and “Gruma” coarse flour, in a proportion of 20, 30, and 50%, respectively. In addition, the different BBH were incorporated into the NCF mix for each treatment.

### 2.2. Physicochemical Analysis of Raw Materials

The proximal composition was made in NCF and BBH. Briefly, samples were milled and perfectly homogenized for the chemical analysis. Moisture content was assessed by the gravimetric method and crude protein was determined by micro Kjeldahl using AOAC methods (925.10 and 960.52, respectively) [15,16]. Fat was determined by the ether extraction method employing Goldfish equipment [17], and ash following the official method AOAC 923.03 [18]. Finally, carbohydrates were calculated as follows:

Carbohydrates by difference:100 − (% Moisture + % Protein + % Ash + % Fat)(1)

(a)Digestible and resistant starch and total dietary fiber.

The digestible and resistant starch contents were determined according to the Megazyme enzymatic protocol (Megazyme, Wicklow, Ireland), K-DST. For K-DST, fractions of rapid, slow, and total digestible were analyzed. Additionally, total dietary fiber was assessed using K-TDFR-200A (Megazyme, Wicklow, Ireland).

(b)Total anthocyanins

Anthocyanins were extracted according to the method previously reported by [19]. After the seed hulls were milled, 3 g was added in 25 mL of a mixture of solvents (acetone/water/acetic acid (70:29.5:0.5, *v*/*v*/*v*)). Then, the mixture was centrifuged twice at 4000 rpm and 4 °C for 20 min. On both occasions, the extract (supernatant) was recovered. Next, the total anthocyanins in tostadas were determined by a pH differential method with potassium chloride buffer (pH 1.0) and sodium acetate buffer (pH 4.5, according to [20]). The absorbance was read at 515 nm (A_λvis-max_) and 700 nm (A_700_) with both buffers. This process was performed in triplicate for each sample. For diluted samples, the net absorbance (A_net_) was calculated with the following formula:A_net_ = (A_λvis-max_ @pH 1.0 − A_700_ @pH 1.0) − (A_λvis-max_ @pH 4.5 − A_700_ @pH 4.5)(2)

Total anthocyanins (TA) were calculated as TA = (A_net_ × MW × DF × 1000)/(ε × 1), where MW = molecular weight, DF = dilution factor, and ε = molar absorptivity. For this study, the predominant anthocyanin in the sample was considered cyanidin-3-glucoside, and the values for MW and ε were 449.2 and 26,900, respectively.

(c)Trypsin inhibitor

Analysis of trypsin inhibitors was executed according to the official method Ba-12a 2020 [21], using N-benzoyl-dl-arginine-p-nitroanilide hydrochloride (BAPNA), with trypsin as the substrate. Values were expressed as trypsin inhibitor units (TIU, or the decrease of 0.01 units with a control trypsin reaction at 410 nm).

(d)Granulometry analysis

Particle size distribution was determined with a laser particle size analyzer (Mastersizer 2000E, Malvern Panalytical, Malvern, UK) coupled with a dry powder feeder (Scirocco 2000M, Malvern Panalytical, Malvern, UK). Standard percentiles of 50, 90, and 98 were considered for each sample. A portion of the sample was added to calibrate the system. Subsequently, three measurements were made, and the average value was reported, employing the Malvern Panalytical Software (Malvern Panalytical, Malvern, UK).

(e)*In vitro* protein digestibility

The in vitro protein digestibility was estimated according to the protocol of [22]. Briefly, 20 mL of a solution of 6.25 mg of protein/mL was adjusted to pH 8 and an enzymatic solution composed of trypsin (T4799, Sigma Aldrich, St. Louis, MO, USA), chymotrypsin (C4129, Sigma Aldrich, St. Louis, MO, USA), and protease of *S. griseus* (P5147, Sigma Aldrich) was added (5 mL). After 10 min, pH was measured, and % digestibility was calculated with the following formula reported by Hsu et al. [22]:% of digestibility = 210.46 − 18.1 × (pH after 10 min).(3)

### 2.3. Tostada Production

NCF was substituted with 10, 15, and 20% BBH by weight for each treatment. The flours were hydrated with tap water until they formed a malleable dough (∼50% moisture). Afterward, the dough was formed and cooked using a commercial tortilla machine with a sheeter device (Villamex Model/V25). The machine was adjusted to create a circular shape with a total weight of approximately 15 g and a thickness of 2 mm. The residence time and temperature were set at 52 s and 280–300 °C, respectively. Tortillas were separated until reaching room temperature, placing them in molds and baking them at 140 °C for 6.4 min.

### 2.4. Physicochemical Analysis of Corn Tostadas

(a)Proximal analysis of corn tostadas.

The proximal composition was made in nixtamalized corn tostadas with black bean hulls (BBH) addition. First, the tostadas were milled in a blender until fine particles were obtained. Then, moisture, crude fiber, ash, fat, and free nitrogen extract determinations were assessed using the same methods used in raw materials.

(b)Digestible and resistant starch, and total dietary fiber.

The digestible, resistant starch, and total dietary were assessed using the Megazyme enzymatic protocols (Megazyme, Wicklow, Ireland), K-DST and K-TDFR-200A.

(c)Predicted Glycemic Index

*In vitro* starch hydrolysis was conducted according to the method used by [23] with modifications. Briefly, 1.0 ± 0.01 g samples were grounded and homogenized in 20 mL of distilled water for a later dispersion in 10 mL of 0.1 M KCl–HCl buffer (pH 1.5) with 200 μL of pepsin solution, with an incubation of 60 min at 37 °C under constant shaking. Following this, 20 mL of maleate buffer (pH 6.2) and 2.5 mL of an enzymatic mixture of pancreatic a-amylase (40 KU/g) and amyloglucosidase (17 KU/g) (PAA/AMG) (Megazyme, Ireland) were added and incubated at 37 °C while shaking. The hydrolysis curve was performed at 30, 60, 90, 120, 150, and 180 min of incubation at 37 °C, 120 rpm, taking 1 mL of the incubated mixture and adding it to capped tubes with 20 mL of 50 mM acetic acid. Next, an aliquot of 2 mL was centrifuged at 13,000 rpm, and 100 μL of the supernatant was mixed with 100 μL amyloglucosidase (100 U/mL) (Megazyme, Ireland), incubating the mixture at 50 °C for 30 min. The glucose content was measured using a GOPOD kit as described for the total and resistant starch procedures. The hydrolysis index (HI) was calculated according to the equation established by [23], dividing the area under the curve (AUC) of each test sample by a white bread curve. The pGI was estimated from the HI value and with the formula proposed by [23]:pGI = 8.198 + 0.862 × HI(4)

(d)Thermo-mechanical behavior of doughs

The thermo-mechanical behavior of control and composite nixtamalized flours substituted with 10%, 15%, and 20% *w*/*w* of BBH was studied using the Mixolab (Mixolab 2, Chopin Technologies, Nanterre, France) and the protocol reported by [24] but using 85 g dough based on preliminary tests. Water absorption (WA) was set at 80% for all samples (∼50% dough moisture). Parameters calculated from the Mixolab curve were: C1: initial consistency; stability: time during which torque was C1-10%; C2: minimum torque during heating from 30 °C to 60 °C; C3: peak torque during heating from 60 °C to 90 °C; gelatinization rate (β): slope during the heating period; C4: minimum torque while holding at 90 °C; C5: torque obtained after cooling to 50 °C [25]. Mixolab analyses were performed in duplicate using distilled water.

(e)Qualitative characterization of anthocyanin profile

The HPLC analysis was carried out with a HPLC-PDA system (1200 Series, Agilent Technologies, Santa Clara, CA, USA) according to [26] with slight modifications. Briefly, 2 μL of resuspended samples in acidified methanol were injected, and the separation was achieved with a Zorbax SB-Aq column (3.0 × 130 mm, Agilent Technologies, Santa Clara, CA, USA). The mobile phases were (A) HPLC-grade water with 0.1% formic acid and (B) HPLC-grade acetonitrile. The flow rate and detector temperature were set at 0.5 mL/min and 45 °C, respectively, with reverse gradient flow. UV–vis absorption spectra were recorded for the predominant peaks at 520 nm. The anthocyanin profile identification was made without a standard, only with the UV–vis spectrum, and by comparison with the previously analyzed patterns [27,28].

(f)Trypsin inhibitor

Analysis of trypsin inhibitors was executed according to the official method Ba-12a 2020 of the AOCS (Chicago, IL, USA), the same as described in Section 2.1 [21].

(g)*In vitro* protein digestibility

The in vitro protein digestibility was estimated according to the protocol of [22] and as briefly described in Section 2.2.

(h)Color

Corn tostada color was analyzed using the Colorimeter X app (v.1.6.6.5, Sao Carlos, BR, USA). All measurements were performed under the same conditions. Briefly, samples were placed on a white surface at a 5 cm distance from the focalized illumination of 45°, 100 Lumens, with a 64 MP obturer. At least six aleatory color measurements were made for each sample, with a previous blank measurement (white surface). The average was used to obtain a color index (CI*). CI* was calculated as [29]
CI* = (a* × 1000)/(L* × b*)(5)
where parameters L*, a*, and b* correspond to lightness, redness, and yellowness, respectively, according to the CIELAB color system. The color difference was calculated as [30].
ΔE = [(ΔL*)^2^ + (Δa*)^2^ + (Δb*)^2^]^½^(6)

(i)Texture

Texture analysis was undertaken for tostadas stored under controlled relative humidity. The analysis was undertaken on days 1, 5, and 8 of storage using a TA.XT2 texturometer (Texture Analyzer Plus, TA Instruments, Surrey, UK). A 25 mm diameter ball probe and a “pipe” cylinder with an outside diameter of 25 mm, a test speed of 5 mm/s, and 0.5 kg force were used to determine the hardness of the samples. This parameter represents the maximum force of the peak load value required to break the samples according to the time–force curve [31].

At least five replicates were obtained until breaking the tostada, and the average value was reported.

(j)Sensory analysis

Sensory analysis was undertaken one day after the tostadas were made. A total of 30 non-trained panelists evaluated treatments, including control, and samples were presented to consumers in a randomized order. Each panelist considered four parameters: color, odor, texture, and taste, and they scored the general acceptability of the tostadas according to a 7-point hedonic scale (from 7: like extremely to 1: dislike extremely) [32,33]. The average of each parameter was reported.

(k)Statistical analysis

All experiments were performed in triplicate. Statistics were determined with analysis of variance (ANOVA; Minitab Statistical Software v. 19, Minitab, Ltd., Coventry, UK) guidelines. The means were compared with Tukey’s test (*p* ≤ 0.05).

## 3. Results and Discussion

### 3.1. Raw Material Characterization

#### 3.1.1. Proximal Composition of the Raw Material

The proximal composition of the raw material used to produce tostadas is summarized in Table 1. NCF showed a moisture content of 10.81%, which is higher than the moisture content in BBH. The proximal composition of NCF was similar to that reported by [34]. Additionally, Palacios et al. [35] previously reported moisture, crude protein, fat, and ash values in the range 9.50–10.32%, 8.43–9.08%, 4.23–4.46%, and 1.12–1.34%, respectively, for three different commercial nixtamalized corn flours.

On the other hand, carbohydrates, by difference, were the main components in both raw materials. As expected, NCF presented a higher content (72.73%) than BBH since cereals, comprises approximately 80% of its weight as carbohydrates. In contrast, legumes such as black beans comprise around 65% of carbohydrates of the total weight [36].

Arockianathan et al. [37] previously reported the proximal composition of *Vigna mungo* L. seed coat, and values were 11.03%, 10.07%, 0.46%, 4.87%, 48.67%, and 27.52% for moisture, crude protein, fat, ash, crude fiber, and carbohydrates by difference, respectively. BBH showed fat and ash content similar to values previously mentioned; moisture and crude fiber were lower, at 6.78% and 16.96% in BBH, respectively. In contrast, carbohydrates by difference and crude protein were higher. However, the values in this study contrast with those reported by Akinjayeju et al. [38]. The highest differences found were in moisture content (10.39% to 6.78%), crude protein (19.93% to 14.66%), fat (2.93% to 0.46%), and crude fiber (4.18% to 16.96%); while the content of ash (3.63% to 4.59%) and carbohydrates by difference (58.94% to 56.56%) were similar. Agronomic differences and the dehulling method have an important effect on the BBH composition.

Dietary fiber content corresponds to insoluble and soluble fractions, mainly non-starch polysaccharides [39]. Dietary fiber content in NCF was similar to previous reports [40]. In BBH, previous reports indicate that most dietary fiber corresponds to insoluble fractions [41].

Although the BBH used as raw material for the production of tostadas was derived from a previous anthocyanin extraction process, they presented a valuable content of 282.03 µg/g. In this sense, the BBH could enhance the nutraceutical properties of nixtamalized corn tostadas. Takeoka et al. reported 250 mg/100 g anthocyanins in the seed coat of black beans (*P. vulgaris* L.) [42]. This value was higher than those reported by Mojica et al. [43] for common bean cultivars, reporting values for anthocyanin concentration ranging from 10 to 2500 µg/g coat.

#### 3.1.2. Starch and Protein Content of the Raw Material

The starch and protein information of the raw materials is shown in Table 2. Total digestible starch (TDS), slowly digestible starch (SDS), and rapidly digestible starch (RDS) were higher in nixtamalized corn flour (NCF) than in black bean hulls (BBH). The RS content in NCF was 1.27%, similar to the range values of 0.98–1.80% previously reported by Agama et al. in four nixtamalized corn flours [44]. TDS was significantly different (*p* < 0.05) between NCF and BBH. In vitro protein digestibility was similar in both raw flours (Table 2). The authors of [45] reported a similar value (81.6%) for protein digestibility in regular nixtamalized maize flour. A different case was observed in trypsin inhibitors, where NCF showed 2.92 TIU/mg, which was lower than BBH. Although trypsin inhibitors are commonly associated with antinutrients, Oliveiria de Lima et al. (2019) highlighted that these molecules are related to appetite control via cholecystokinin metabolism [46].

The particle diameter showed significant differences (*p* < 0.05) among samples. NCF had a higher particle diameter due to the mixture of nixtamalized flours with different particle sizes, which is desirable for maize snack products, according to Serna-Saldivar et al. [47]. The values ranged between 281.3–876.4 µm and 78.9–501.9 µm for NCF and BBH, respectively. BBH showed a lower diameter in percentile 50 with a particle diameter of 78.93 µm, attributed to the way it was grounded.

#### 3.1.3. Thermo-Mechanical Behavior of Doughs

The results of the thermo-mechanical behavior of the control and composite nixtamalized flours evaluated with the Mixolab are summarized in Table 3.

C1 represents the maximum consistency of doughs during the initial mixing stage of Mixolab analysis. C1 significantly increased with increasing levels of BBH. Since the WA was fixed at 80% for all samples, the higher initial consistency of dough supplemented with BBH was associated with a competition for water among NCF and BBH components. BBH contained a significantly higher crude fiber content than NCF (Table 1). Due to the presence of hydroxyl groups in the fiber structure, which enable more interactions with water through hydrogen bonds [48], a higher WA would be needed in doughs containing BBH to obtain the same consistency as the control. This could benefit the production yield of tortillas enriched with BBH, especially if the water is retained after baking. Stability was lower in doughs substituted with 10% and 15% BBH compared with the 100% NCF control. Meanwhile, substitution with 20% BBH did not significantly affect this parameter (Table 3). Stability time has been positively correlated with the particle size of NCF and negatively correlated with its gelatinization degree and leached amylose [24,25]. The lower particle sizes of BBH flour (Section 3.1.2) could explain the reduction in stability time observed in doughs containing this raw material. On the other hand, the higher stability of the dough containing 20% BBH, compared to the counterparts with 10% and 15%, could be associated with the lower amount of pregelatinized starch contributed by NCF. Stability is a relevant parameter during processing. Nixtamalized dough with low stability time becomes sticky and adhesive during mixing, negatively affecting its machinability [24,25].

C2 values did not present significant differences even when C1 values differed among samples, resulting in greater C1–C2 values in samples with higher BBH levels (Table 3). The C1–C2 decline indicates the weakening of doughs during initial heating, and higher values have been correlated with finer particles in NCF and lower fat content [24]. This is consistent with the findings in this study, where increasing levels of BBH with finer particles led to composite flours with lower fat content (Table 4). C3 values, which are associated with starch gelatinization, were higher in the control dough compared to dough supplemented with BBH. The starch content of BBH diluted that of NCF and lowered C3 values in composite doughs. The gelatinization rate (β) was also reduced by adding BBH dough (Table 3). A previous study suggested that higher β values could indicate a quicker water uptake and swelling of starch granules [25]. Therefore, the reduced β values were associated with the dilution of starch and the presence of molecules in BBH that restrained proper hydration and swelling of starch granules. C4 was also reduced with increasing levels of BBH substitution. Higher C4 values are related to lower gel cooking stability [25], which could also be related to starch dilution in doughs containing BBH. Finally, the BBH addition did not significantly affect the C5 values related to starch retrogradation.

### 3.2. Corn Tostadas Characterization

#### 3.2.1. Proximal Composition of Corn Tostadas

As can be seen in Table 4, adding 20% BBH to tostadas increased protein significantly (*p* < 0.05). Values were in the range reported previously by Grajales et al. [39] in tortillas blended with black bean flour. Samples with 10% and 15% addition did not differ significantly from the control sample. The most significant difference found between the treatments was in ash, carbohydrates by difference, and dietary fiber. The increase in ash content for tostada with 20% BBH was related to the mineral content of legumes as previously reported [49]. Legumes greatly contribute minerals [50]. Tostadas with BBH had a similar fat content than NCF, these were in the range of 0.44 to 1.55%. Compared to the control, tostadas with different BBH additions had lower values of carbohydrates by difference, going from 81.72% to 76.08%, without differences between treatments. This reduction was due to the high levels of protein in BBH.

Dietary fiber content was significantly different (*p* < 0.05) in all treatments, with the highest content in tostadas with a 20% addition of BBH. Non-starch polysaccharides, such as cellulase, hemicellulose, and resistant starch, among others, are the main components of dietary fiber in BBH [39,41].

The total anthocyanin content for BBH was 282.03 µg/g. Theoretically, the concentration of anthocyanins in tostadas should be 28.2, 42.3, and 56.4 µg/g for those with 10, 15, and 20%, respectively. This corresponds with the results shown 2 months after the tostada elaboration. The presence of anthocyanins in the sample with the highest addition of BBH was also reflected in its color, with the sample with 20% BBH being the one with the greatest difference in terms of this parameter compared to the control. These are similar to those previously reported by [51].

#### 3.2.2. Starch and Protein Information for Corn Tostadas

Starch and protein contents in corn tostadas are summarized in Table 5.

The RDS contents in tostadas with BBH were similar to those of NCF (Table 5). Tostadas with 10% and 15% showed similar SDS content. A similar effect was observed in control tostadas and with the addition of 20% of BBH. On the other hand, TDS was similar between NCF tostadas and those with different percentages of BBH. Treatment with 20% BBH had the highest RS content, which increased from 0.46% to 2.30% compared to control tostadas. This can be explained by the complexes between fatty acids and starch granules (RS type 5) in baking [52]. The predicted glycemic index (pGI) was estimated by starch hydrolysis for each treatment. The values obtained correspond with the pGI reported for common nixtamalized maize tortillas, with values around 50–60. A clear correlation was shown between BBH addition and lower pGI, although it was slightly different between the BBH treatments. Adding BBH in higher concentrations may induce a further reduction in glycemic response, which is related to the dietary fiber content in BBH. Similar findings were previously reported for tortillas mixed with whole black bean flour [39].

*In vitro* protein digestibility and trypsin inhibitors ranged from 84.22–85.57, and 6.10–8.86 TIU/mg, respectively, and these values did not show significant differences (*p* ≤ 0.05) between treatments. Mora et al. [45] previously evaluated the protein digestibility behavior in tortillas of regular and commercial nixtamalized maize flour; the values were 82.8% and 82.0%, respectively.

#### 3.2.3. Qualitative Characterization of Anthocyanin Profile by HPLC for Corn Tostadas

The chromatographic analysis showed two main glycosylated anthocyanins, cyanidin and malvidin (Figure 1), in the 20%-enriched nixtamalized corn tostadas. Similar results have been previously reported [27,53]. Hsieh-Lo et al. [53] identified the same anthocyanin profile in a black bean anthocyanin-rich extract by ultra-high-performance liquid chromatography (UHPLC). However, it should be noted that differences in the flavonoid profile are directly related to the bean variety used, as reported previously [30]. Cyanidin, which is a type of flavonoid molecule, and a member of the anthocyanins present in black beans, is associated with health benefits. Several studies have suggested that it can improve obesity and type 2 diabetes treatments [54]. On the other hand, Mojica et al. [30] reported the potential of purified anthocyanins, delphinidin and malvidin, from black bean anthocyanin-rich extracts as inhibitors of enzymes and their use as molecular targets for diabetes.

Flavonoids were detected in the final product after the toasting process. Further research can evaluate their quantification and functionality in the toasts.

#### 3.2.4. Color Analysis of Corn Tostadas

Color is a quality parameter with special importance for consumer acceptance [49]. In this study, we evaluated the color change of tostadas with different levels of BBH addition. The results of the color analysis are summarized in Table 6. The L* parameter corresponds to luminosity, which was higher in control tostadas. L* values decreased as BBH inclusion increased, with tostadas with 20% BBH addition presenting the lowest values due to the characteristic dark color of BBH. However, tostadas with 10% and 15% BBH presented similar L* values. The a* values indicate the variation between red and green in the spectrum. Treatments did not show significant differences (*p* ≤ 0.05) in a*, and higher values were reported in control tostadas. The negative *a** values in tostadas with BBH were related to a reddish hue, while control tostadas presented a greenish hue. In the same way, b* parameters are related to the variation between yellow and blue in the spectrum. In this case–control and 10%, BBH tostadas showed similar values and the same pattern in tostadas with 15% and 20% BBH. Previous reports showed similar values in L* and a* parameters (48.78 and −0.29, respectively) in nixtamalized maize tortillas with 3 g/kg black bean extract to those obtained in tostadas with 10% BBH addition [55].

The color index (CI*) is a quality parameter in food evaluation. Positive values in CI* ranging from 20 to 40 are related to the color variation between orange and dark red. Control tostadas showed a CI* of 33.39, corresponding to the previously mentioned color characteristics. Tostadas with 10% BBH had a CI* of −1.82, representing a greenish-yellow color. On the other hand, tostadas with 15–20% BBH inclusion reported negative CI* values of −3.72 and −2.97, respectively, which are related to deep green and yellowish green tones. Additionally, ΔE* represents the change in color regarding control tostadas [30]. These values significantly increased with increasing levels of BBH. The ΔE* values ranged from 39.21 to 55.96, and differences were not shown between tostadas with 10% and 15% BBH inclusion.

Figure 2 shows the final samples according to the BBH addition. As the BBH inclusion increased, the blackness increased, as shown in tostadas with 20% BBH addition (D) concerning the control (A).

#### 3.2.5. Hardness Analysis of Corn Tostadas

Figure 3 shows the hardness results of nixtamalized corn tostadas during 8 days of storage. BBH addition affected the tostadas hardness. The hardness element is crucial for developing crispy food products and is directly related to their general acceptability [56]. The results showed an increase in hardness by increasing the BBH addition on day one of storage and a reduction on day 5. However, a slight difference was observed in tostadas with 15% BBH, which increased. On day 8, 10% and 15% tostadas showed similar hardness.

In contrast, control tostadas showed the lowest change in hardness during storage. Some factors impacting the hardness changes could be the ratio of fiber and protein associated with the BBH addition to tostadas. Finally, ref. [31] assessed that baking time, due to small air cells that expand and are joined together by the gelatinized starch during the baking process, also affects the textural properties of food.

#### 3.2.6. Sensory Analysis of Corn Tostadas

Figure 4 shows the results of the sensory evaluation of tostadas. Panelists evaluated four parameters and the general acceptability of tostadas with other BBH additions. Generally, the tostada with 15% of BBH was the best-rated after the control, and the sample with 20% inclusion was the worst, according to a hedonic scale of 7 points. Samples with 10% and 20% inclusions were evaluated as similar. Panelists evaluated the color parameter similarly in all samples, and the ratings were presented in a range of 60.89 to 62.15. The odor parameter is considered an essential sensory parameter in organoleptic characteristics [49]. The tostada with 20% of the BBH addition showed the least acceptance in this parameter, and values decreased from 62.33 to 60.91 in contrast with the control sample. A similar pattern was observed in texture, taste, and general acceptability because tostadas with 15%, 10%, and 20% BBH were evaluated in the same order from most to least acceptance in the three parameters.

## 4. Conclusions

The addition of black bean hulls (BBH) in nixtamalized corn tostadas (at 10%, 15%, and 20%) resulted in an increase in crude protein (from 8.65% to 10.93%), crude fiber (from 3.30% to 6.00%), and dietary fiber (from 3.15% to 5.65%) compared to control tostadas enriched with 20% BBH. Moreover, the starch fraction analysis showed a decrease in total digestible starch (from 61.97% to 59.07%), an increase in resistant starch (from 0.46% to 2.3%), and a decrease in the predicted glycemic index (from 52 to 49), among other parameters. These results suggest that BBH is a suitable alternative for developing nutraceutical food products. The chromatographic analysis showed the presence of two main glycosylated anthocyanins: cyanidin and malvidin. We also observed improvements in the physicochemical, textural, and sensory characteristics of the tostadas, which further supports BBH as a suitable choice for developing nutraceutical products. Further research should be conducted to investigate the use of BBH as a functional ingredient and its potential as a byproduct.

## Figures and Tables

**Figure 1 foods-12-01915-f001:**
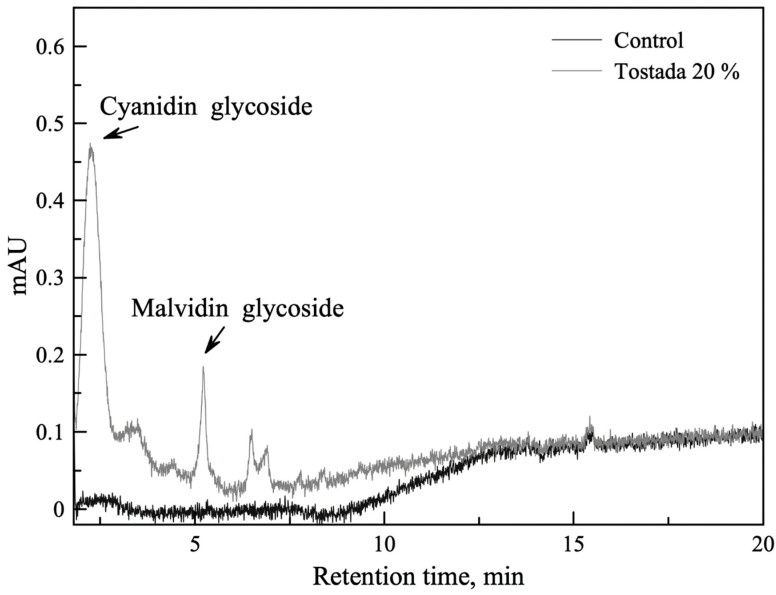
Detection of the presence of glycosylated anthocyanins in 20%-enriched nixtamalized corn tostadas.

**Figure 2 foods-12-01915-f002:**
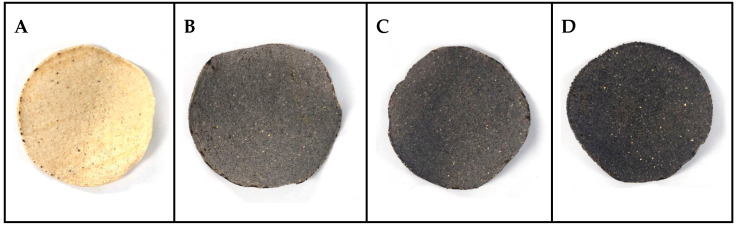
(**A**) The control sample, (**B**) Tostada 10% BBH, (**C**) Tostada 15% BBH, and (**D**) Tostada 20% BBH addition.

**Figure 3 foods-12-01915-f003:**
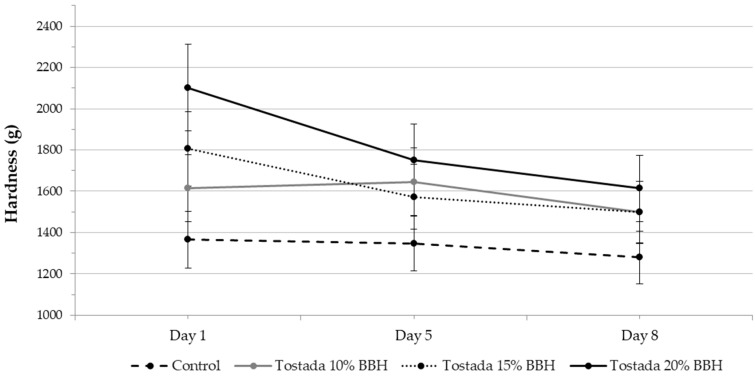
The effect of black bean hulls on the hardness (g) of nixtamalized corn tostadas during 8 days of storage.

**Figure 4 foods-12-01915-f004:**
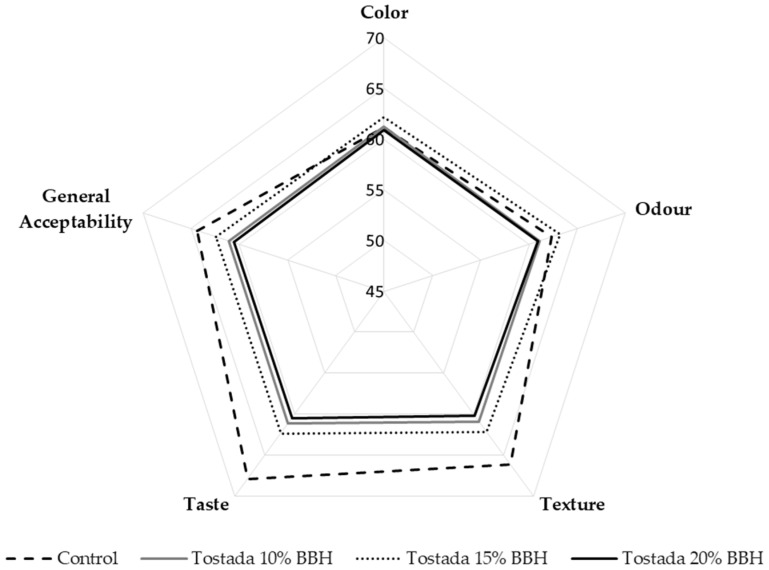
The effect of black bean hulls on the sensory properties of nixtamalized corn tostadas—color, odor, texture, taste, and general acceptability.

**Table 1 foods-12-01915-t001:** Proximal composition of nixtamalized corn flour and black bean hulls as raw material to produce tostadas ^1^.

Flour	Moisture (%)	Crude Protein (%)	Fat (%)	Ash (%)	Crude Fiber (%)	Carbohydrates by Difference (%)	Dietary Fiber (%)	Anthocyanins (µg/g)
Nixtamalized corn flour (NCF)	10.81 ± 0.12 ^a^	8.34 ± 0.19 ^b^	1.87 ± 0.40 ^a^	1.50 ± 0.01 ^b^	4.76 ± 1.00 ^b^	72.73 ± 1.58 ^a^	6.26 ± 0.10 ^b^	-
Black bean hulls (BBH)	6.78 ± 0.19 ^b^	14.66 ± 1.07 ^a^	0.46 ± 0.02 ^b^	4.59 ± 0.05 ^a^	16.96 ± 0.50 ^a^	56.56 ± 1.17 ^b^	10.55 ± 0.15 ^a^	282.03 ± 25.15

^1^ All results are on a dry basis and are the average of at least three replicas ± standard deviation. Different superscripts within a column indicate statistical difference (*p* < 0.05).

**Table 2 foods-12-01915-t002:** Starch, protein, and particle size data for nixtamalized corn flour and black bean hulls (BBH) as raw materials to produce tostadas ^1^.

Flour	Total Digestible Starch (TDS, %)	Slowly Digestible Starch (SDS, %)	Rapidly Digestible Starch (RDS, %)	Resistant Starch (RS, %)	*In Vitro* Protein Digestibility	Trypsin Inhibitor (TIU/mg)	Average Particle Size (µm)		
							Percentile		
							50th	90th	98th
Nixtamalized corn flour (NCF)	53.7 ± 3.2 ^a^	21.8 ± 0.3 ^a^	20.8 ± 1.1 ^a^	1.2 ± 0.1 ^b^	83.5 ± 1.3 ^a^	2.9 ± 0.2 ^b^	281.3 ± 4.7 ^a^	698.5 ± 7.5 ^a^	876.4 ± 3.1 ^a^
Black bean hulls (BBH)	16.8 ± 0.9 ^b^	1.5 ± 0.1 ^b^	3.5 ± 0.2 ^b^	7.6 ± 0.6 ^a^	76.6 ± 4.1 ^a^	7.4 ± 0.2 ^a^	78.9 ± 1.0 ^b^	309.5 ± 1.6 ^b^	501.9 ± 5.6 ^b^

^1^ All results are on a dry basis and are the average of at least three results ± standard deviation. Different superscripts within a column indicate statistical difference (*p* < 0.05).

**Table 3 foods-12-01915-t003:** Mixolab parameters of doughs produced with nixtamalized corn flour and black bean hulls ^1^.

Parameter	Control	Tostada 10% BBH	Tostada 15% BBH	Tostada 20% BBH
C1 (Nm)	1.95 ± 0.02 ^d^	2.14 ± 0.01 ^c^	2.22 ± 0.01 ^b^	2.37 ± 0.01 ^a^
Stability (min)	12.95 ±0.49 ^a^	10.55 ± 0.21 ^b^	10.85 ± 0.21 ^b^	11.5 ± 0.57 ^ab^
C2 (Nm)	1.16 ± 0.06 ^a^	1.24 ± 0.03 ^a^	1.27 ± 0.07 ^a^	1.3 ± 0.01 ^a^
C3 (Nm)	2.08 ± 0.05 ^a^	1.95 ± 0.01 ^b^	1.85 ± 0.01 ^b^	1.83 ± 0.04 ^b^
Gelatinization rate (β)	0.35 ± 0.05 ^a^	0.25 ± 0.02 ^ab^	0.17 ± 0.01 ^b^	0.17 ± 0.06 ^b^
C4 (Nm)	1.96 ± 0.08 ^a^	1.83 ± 0.02 ^ab^	1.75 ± 0 ^b^	1.72 ± 0.01 ^b^
C5 (Nm)	2.98 ± 0.19 ^a^	2.79 ± 0.01 ^a^	2.82 ± 0.02 ^a^	2.84 ± 0.02 ^a^

^1^ Values are the means and standard deviations of two replicates. Averages with the same letter within the same parameter (row) are not significantly different at *p* < 0.05. Abbreviations: Control, 100% nixtamalized corn flour; 10–20% BBH, composite nixtamalized corn flours with 10–20% BBH; C1, initial consistency; stability, the time during which torque was C1-10%; C2, minimum torque during heating from 30 °C to 60 °C; C3, peak torque during heating from 60 °C to 90 °C; gelatinization rate (β), slope during the heating period; C4, minimum torque during holding at 90 °C; C5, torque obtained after cooling to 50 °C.

**Table 4 foods-12-01915-t004:** Proximal composition of corn tostadas ^1^.

Sample	Nixtamalized Corn Flour (%)	Black Bean Hulls (%)	Moisture (%)	Crude Protein (%)	Fat (%)	Ash (%)	Crude Fiber (%)	Carbohydrates by Difference (%)	Dietary Fiber (%)	Total Anthocyanin (µg/g Sample) Time: 2 Months
Control	100	-	3.77 ± 0.20 ^c^	8.65 ± 0.20 ^b^	1.06 ± 0.05 ^a,b^	1.49 ± 0.00 ^d^	3.30 ± 0.90 ^b^	81.72 ± 1.08 ^a^	3.15 ± 0.17 ^d^	-
Tostada 10% BBH	90	10	4.93 ± 0.04 ^a^	9.26 ± 0.50 ^b^	1.55 ± 0.37 ^a^	1.82 ± 0.02 ^c^	4.66 ± 0.90 ^a,b^	77.78 ± 1.37 ^b^	4.02 ± 0.2 ^c^	28.20 ± 2.51 ^b^
Tostada 15% BBH	85	15	4.37 ± 0.04 ^b^	9.37 ± 0.71 ^b^	0.71 ± 0.30 ^b^	2.00 ± 0.01 ^b^	5.12 ± 1.10 ^a,b^	78.43 ± 1.77 ^b^	4.79 ± 0.33 ^b^	55.48 ± 5.17 ^a^
Tostada 20% BBH	80	20	4.40 ± 0.02 ^b^	10.93 ± 0.58 ^a^	0.44 ± 0.32 ^b^	2.15 ± 0.01 ^a^	6.00 ± 0.30 ^a^	76.08 ± 0.25 ^b^	5.65 ± 0.40 ^a^	66.61 ± 10.34 ^a^

^1^ All results are on a dry basis and are the average of at least three results ± standard deviation. Averages with the same letter within the same column are not different at *p* < 0.05.

**Table 5 foods-12-01915-t005:** Starch and protein information for corn tostadas ^1^.

Sample	Nixtamalized Corn Flour (%)	Black Bean Hulls (%)	Rapidly Digestible Starch (RDS, %)	Slowly Digestible Starch (SDS, %)	Total Digestible Starch (TDS, %)	Resistant Starch (RS, %)	Predicted Glycemic Index	*In Vitro* Protein Digestibility	Trypsin Inhibitor (TIU/mg)
Control	100	-	37.47 ± 0.96 ^a^	22.21 ± 0.58 ^a^	61.97 ± 1.91 ^a,b^	0.46 ± 0.01 ^c^	52.024 ± 0.14 ^a^	85.57 ± 2.70 ^a^	6.76 ± 0.87 ^a^
Tostada 10% BBH	90	10	36.17 ± 3.30 ^a^	13.10 ± 0.79 ^b^	65.52 ± 4.52 ^a^	1.25 ± 0.08 ^b^	50.96 ± 0.07 ^b^	85.33 ± 2.89 ^a^	8.86 ± 1.90 ^a^
Tostada 15% BBH	85	15	36.04 ± 0.94 ^a^	13.65 ± 0.43 ^b^	56.27 ± 3.66 ^b^	1.14 ± 0.08 ^b^	50.62 ± 0.07 ^b^	85.51 ± 1.38 ^a^	6.10 ± 0.25 ^a^
Tostada 20% BBH	80	20	35.73 ± 1.16 ^a^	20.94 ± 2.21 ^a^	59.07 ± 2.79 ^a,b^	2.30 ± 0.20 ^a^	49.17 ± 0.13 ^c^	84.22 ± 0.57 ^a^	7.98 ± 0.88 ^a^

^1^ All results are on a dry basis and are the average of at least three results ± standard deviation. Averages with the same letter within the same column are not different at *p* < 0.05.

**Table 6 foods-12-01915-t006:** The effect of black bean hulls (BBH) on the color parameters of nixtamalized corn flour tortillas ^1^.

Parameter	Nixtamalized Corn Flour (100%)	Nixtamalized Corn Flour (90%): Black Bean Hulls (10%)	Nixtamalized Corn Flour (85%): Black Bean Hulls (15%)	Nixtamalized Corn Flour (80%): Black Bean Hulls (20%)
L*	85.10 ± 3.99 ^a^	47.00 ± 5.34 ^b^	41.37 ± 3.94 ^b^	29.87 ± 2.67 ^c^
a*	8.61 ± 0.81 ^a^	−0.25 ± 0.12 ^b^	−0.25 ± 0.21 ^b^	−0.13 ± 0.14 ^b^
b*	3.05 ± 0.41 ^a^	3.67 ± 1.28 ^a^	1.72 ± 0.76 ^b^	1.43 ± 0.25 ^b^
Color Index	33.39 ± 2.37 ^a^	−1.82 ± 1.51 ^b^	−3.72 ± 3.10 ^b^	−2.97 ± 2.51 ^b^
Delta E	-	39.21 ± 8.23 ^b^	44.65 ± 3.1 ^b^	55.96 ± 4.92 ^a^

^1^ All results are the average of at least three results ± standard deviation. Averages with the same letter within the same row are not different at *p* < 0.05.

## Data Availability

The data presented in this study are available on request from the corresponding author. The data are not publicly available due to privacy restrictions.

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
