# Peer review of "Black Bean Hulls as a Byproduct of an Extraction Process to Enhance Nutraceutical and Glycemic-Related Properties of Nixtamalized Maize Tostadas"

_foods, 2023, doi:10.3390/foods12091915_

Round 1

Reviewer 1 Report

The reviewed manuscript is interesting, however, in its current form it has a number of errors, especially in the field of sensory, chromatographic and color evaluation.

Detailed comments are given below.

Line numbers refer to the pdf version of the manuscript.

Introduction: In my opinion, there is a need to explain the nixtamalization technique to non-Latin American audiences. Two or three sentences needed.

Materials and Methods:

Line 86. Lack of dot and space.

Line 85 and 86 Wouldn't it be better to arrange the values from least to greatest? Btw in line 142 and 184 other percentages are given. Please explain.

Line 105 Please check the degree symbol

Line 140 Please check the multiplication symbol

Line 147 and 148 Please check the degree symbol

Line 95-100 and 158-161 Please provide more details about the analysis.

Line 164 remove the minus sign in front of the grams symbol

Line 167; 170; 171;176 Please check the degree symbol

Line 174 remove the minus sign in front of the micro liters symbol

Line 202 Please check the degree symbol

Line 195 – 206 Please provide detection limits (LOD and LOQ) for individual compounds

Line 234-241 Please provide the approval number of the ethics committee for sensory testing analysis. Currently, publishing standards require Authors to provide appropriate consents.

Line 278-282 Individual fractions of dietary fiber have not been marked, so a discussion on this subject is not justified.

Line 287 remove the minus sign in front of the grams symbol

Table 3 Please pay attention to the formatting of the table

Line 345-348 Why does a 10% addition of BBH lower the stability, while increasing the additive brings the stability back to the standard level?

Table 4 Please pay attention to the formatting of numerical values in table 4

Section 3.2.3 I have strong reservations about the chromatographic analysis. The presented chromatograms show a lot of noise. The peaks identified by the authors are probably below the detection level (peak shape and noise). I believe that these results are not suitable for publication. The analysis should be repeated in such a way as to obtain a strong analytical signal.

Line 468 The "a" symbol is missing

All tables: Please unify the sample designations in the tables (e.g. 10% BBH; 15% BBH; 20% BBH).

Line 527-529 How is it possible that in the sensory analysis samples that differ so significantly in color received a similar evaluation in terms of this parameter? The fact that the sensory panel does not show differences in color is puzzling. Please provide statistics for sensory analysis

The article should be checked by a native speaker.

Author Response

Dear reviewer, thanks for your time and feedback. Our punctual responses are in the attached file. Best, 

Reviewer 2 Report

The manuscript titled "Black bean hulls as a byproduct of an extraction process to enhance nutraceutical and glycemic-related properties of nixtamalized maize tostadas" deals within the scope of the Foods Journal, by investigating an interesting topic of research. However, reading this article raises several questions and objections.

Major remarks:

Based on the proximate composition of the black bean hulls given in Table 1, it appears that the material used contains not only the hull (husk) but also other parts of the bean kernel. This is further supported by the fact that the carbohydrate and ash contents reported in this research are comparable to those reported by Akinjaye and Ajayi (Ref. 36; Lines 271-277), where they presented the chemical composition of the entire beans, not just the husks. Additionally, in the Materials & Methods section, the authors only state that the bean hull was obtained as a byproduct in the anthocyanin extraction process (Line 82), without providing a detailed description of how the bean husk was obtained and how it was ensured that this material contains only the bean husk and not other residual parts of the bean kernel.

According to the Foods journal Instructions for authors " Materials and Methods should be described with sufficient detail to allow others to replicate and build on published results." Therefore, the authors should provide a more detailed description of the method used to obtain the black bean hull to ensure that the methods presented are repeatable by other researchers, and to satisfy one of the key aspects of science: replicability and reproducibility.

Minor remarks:

Lines 63-67: Please provide references for the statements in this paragraph. Also, the authors should include a sentence or two about the technology of extracting anthocyanins from black bean, since they used the byproduct of this type of process in this research.

Lines 214-222: To ensure the accuracy and reliability of color measurement using a colorimeter, it is important to perform calibration of the colorimeter, standardize the conditions, prepare the samples properly, ensure measurement repeatability, compare the samples to reference values, eliminate interfering factors, and adhere to the measurement protocol. However, in this study, it appears that these preparations were not followed. The manuscript lacked essential information about the camera used, calibration procedure, lighting conditions, position of the samples and camera, etc. Without this information, replicating the experiment would be impossible, which is a fundamental requirement in scientific research.

Line 219: Please, provide the reference for color index equation.

Author Response

Dear reviewer, thanks for your time and feedback. Our punctual responses are in the attached file. Best, 

The authors

Reviewer 3 Report

The study of Machado-Velarde is interesting, but the manuscript required major revision before final decision to be made:

Lines 62-63: The statement “Among these strategies is the …. food supplements” needs revision. The main idea is difficult to follow.

Lines 79 and 88: Sub-sections need renumbering; 2.1 appears twice.

Line 112: Pay attention to the subscripts. Provide the meaning for (the different) A: first you use it as anthocyanin, then as absorbance. The readers might be confused. Please check the entire section for this kind of change.

Line 140: Please provide the meaning of the coefficients used in the digestibility formula.

Lines 142: It is not clear what “NCF was mixed with BBH for 10, 15, and 20% treatments” means. Please revise. It should be clearly indicated that the values refer to substitution ratios.

Lines 145: Tenses of the verbs need revision.

Line 209: Section 2.2 should be recalled instead of 2.1.

In Table 2 it is not clear what is the meaning of 50-90-98. More explicit information should be included in the caption of the head of the table – I think would be better to indicate the percentile together with the values. Only in the text, later on, everything becomes clear.

Table 3: It is useless to provide the values od C1-C2 since you already indicate the values of C1 and C2. Please remove the C1-C2 values from the table.

The same sample codification should be used in Tables 3-5

Line 455: Most probably the authors referred to the “inhibitor of the enzymes” or “enzymes’ inhibitor”. Please revise.

How ere the black bean hulls obtained. More details on the anthocyanin extraction process should be indicated, if available. This information would enable understanding some result presented in the manuscript, especially those related to the presence of anthocyanins and flavonoids (subsection 3.2.3). Moreover, the procedure used for the anthocyanin extraction might significantly influence the Thermo-mechanical behavior of doughs.

Lines 543-545: The statement needs revision. It is not correct to discuss about the increase in the control in respect to the working samples.

The English should be carefully checked in the entire manuscript.

The English should be carefully checked in the entire manuscript.

Author Response

(The authors gave the same response as above.)

Reviewer 4 Report

The work sent for review under the label foods-2354889, is dedicated to the inclusion of black bean husk residues in the production of nyctmalized corn tostada (addition 10, 15 and 20%), and then the physico-chemical, textural and sensory properties of the obtained tostada were examined. The work has good practical and environmental significance, the analyzes are well planned and conducted, the results are interesting, but there are several shortcomings that need to be corrected.

Line 53-60 Delineate which of the compounds listed here are nutritionally desirable and which are considered antinutrients and it is desirable to reduce their content (such as lectin, phytic acid etc).

Line 68 Clarify the process of nixtamalization.

Line 68-77 Look back at whether there are any manuscripts in which black bean hulls are used for food production and emphasize what is the main innovation of this manuscript.

Line 94 Write the equation for calculating carbohydrates to make it clearer and label that equation with the number 1.

Line 95-100 Expand the discussion for these methods.

Line 109, 181, 281, 321, 373, 379 etc. should be checked through the entire text, etc. should be checked through the entire text, Not only the number of the paper should be written, but the author or author et al. if there are more than one.

Line 140 Label this equation with the number 2.

Line 181 Separate the equation into a new line and mark it with number 3.  Same for Line 222.

Line 215 Manufacturer city and state for colorimeter?

Line 242 The part that describes which statistical methods were applied to process the results is missing.

Line 246-251 Compare with the obtained results and try to explain why less fat and protein were obtained.

Line 278-282 Why was no analysis done to determine how much soluble and insoluble dietary fiber there is, to support the claim that most of the fiber in BBH is insoluble.

Line 292-302 It is necessary to describe this part related to starch in more detail, as well as the division into starches according to digestibility and their role.

Line 306-307 Also describe in more detail the importance of trypsin inhibitors and emphasize as a positive advantage of the BBH supplement that it has 2.5 times more content of trypsin inhibitors than NCF.

Line 455 Describe figure  1 and emphasize that the tostada with 20% BBH has the highest concentration of cyanidin and malvidin.

Line 468 „The * values..“ Did you mean a?

Line 528 Panelist evaluated the color parameter similarly in all samples.. how?

Author Response

(The authors gave the same response as above.)

Round 2

Reviewer 1 Report

The corrections made are sufficient.

Reviewer 2 Report

The authors have made all requested changes and improvements.

Reviewer 3 Report

The manuscript was improved and can be accepted for publication.

Minor English improvement could be applied.